# Graphene Oxide Strengthens Gelatine through Non-Covalent Interactions with Its Amorphous Region

**DOI:** 10.3390/molecules29112700

**Published:** 2024-06-06

**Authors:** Hak Jin Sim, Katarina Marinkovic, Ping Xiao, Hui Lu

**Affiliations:** 1School of Biological Sciences, Faculty of Biology, Medicine and Health, The University of Manchester, Manchester M13 9PT, UK; simhakjin@hotmail.com (H.J.S.); katarina.marinkovic@postgrad.manchester.ac.uk (K.M.); 2Department of Materials, Faculty of Science and Engineering, The University of Manchester, Manchester M13 9PL, UK; p.xiao@manchester.ac.uk; 3Henry Royce Institute, The University of Manchester, Manchester M13 9PL, UK

**Keywords:** graphene oxide, 2D nanomaterials, gelatine, cytotoxicity, mechanical property of gelatine, biomaterials

## Abstract

Graphene oxide (GO) has attracted huge attention in biomedical sciences due to its outstanding properties and potential applications. In this study, we synthesized GO using our recently developed 1-pyrenebutyric acid-assisted method and assessed how the GO as a filler influences the mechanical properties of GO–gelatine nanocomposite dry films as well as the cytotoxicity of HEK-293 cells grown on the GO–gelatine substrates. We show that the addition of GO (0–2%) improves the mechanical properties of gelatine in a concentration-dependent manner. The presence of 2 wt% GO increased the tensile strength, elasticity, ductility, and toughness of the gelatine films by about 3.1-, 2.5-, 2-, and 8-fold, respectively. Cell viability, apoptosis, and necrosis analyses showed no cytotoxicity from GO. Furthermore, we performed circular dichroism, X-ray diffraction, Fourier-transform infrared spectroscopy, and X-ray photoelectron spectroscopy analyses to decipher the interactions between GO and gelatine. The results show, for the first time, that GO enhances the mechanical properties of gelatine by forming non-covalent intermolecular interactions with gelatine at its amorphous or disordered regions. We believe that our findings will provide new insight and help pave the way for potential and wide applications of GO in tissue engineering and regenerative biomedicine.

## 1. Introduction

Tissue engineering is an interdisciplinary field combining engineering and life sciences to regenerate or enhance the function of damaged tissues and/or whole organs. Such injuries and diseases represent major healthcare challenges [1,2]. Collagen, the main extracellular matrix (ECM) protein, is widely used as a biomaterial for tissue engineering scaffolds [3,4] However, collagen is costly and insoluble in water, which limits its use in tissue engineering studies [5]. Gelatine, derived from collagen through hydrolysis, is widely used in cell culture and tissue engineering for its good biocompatibility, solubility, and gelation ability [6,7]. Moreover, gelatine production is more scalable [8], and it has relatively lower immunogenicity than collagen [9,10]. Thus, gelatine is often used as a substitute for collagen in tissue engineering and regenerative medical studies [11]. On the other hand, a main drawback and limitation of gelatine is its poor mechanical properties [12]. To improve its mechanical properties, reinforcement with fillers or crosslinking, such as multi-walled carbon nanotubes [13] and nanocellulose [14], has been used. However, the presence of residual crosslinking agents and high surface-to-volume ratios of nanofillers tends to induce higher reactivity and toxic side effects [15,16,17,18].

Graphene oxide (GO) is a chemical derivative of graphene, containing many oxygenated functional groups (e.g., hydroxyls, epoxides, ketones, and carboxyls) [19]. GO has attracted tremendous interest in the biomedical field, due to its good biocompatibility, water dispersibility, and chemically modifiable large surfaces. These make GO an increasingly important nanomaterial, which exhibits great promise in the area of nanobiomedicine and tissue engineering [20]. When combined with biopolymers, like gelatine and chitosan, the oxygenated groups in GO can act as interfacial linkers, via hydrogen, ionic, and/or covalent bonds, and facilitate stress transfer from the biopolymer matrix to GO [21,22,23]. This includes ionic interactions between the carboxylic groups of GO and double-stranded DNA [24], π-π stacking between GO and plasmid DNA [25], and covalent bonding between GO and biological enzymes, such as esterification between the amino groups of (+) γ-lactamase enzyme and the epoxy groups of GO [26,27]. Consequently, the addition of GO increases the mechanical properties of subsequent biopolymer composites. For instance, the tensile strength of chitosan increased by 122% when 1 wt% GO was incorporated into chitosan films [28]. Another study showed an increase of over 160% in the tensile strength of a poly(ε-caprolactone) scaffold upon 0.1 wt% GO addition [29]. However, excessive GO in polymers has negative effects on mechanical properties [29,30,31,32]. This seems due to poor GO dispersion, causing GO agglomeration and resulting in inefficient stress transfer between the GO sheets and polymer matrix [33,34]. Thus, the effects of GO in biomaterials are complex, and, thus, more studies are required to assess its potential successful use in tissue engineering and regenerative medicine.

Assessing the in vitro cytotoxicity and safety profile is essential for the development of nanomaterial-based formulations for biomedical applications. Several studies have indicated that GO is biocompatible across various cell lines in vitro [35,36,37,38]. However, there are also conflicting literature reports on the biocompatibility and cytotoxicity of GO [39,40,41,42,43], due to various factors, including GO purity, concentration, lateral size, incubation time, and type of cells used [44,45,46,47]. For example, using HEK-293 cells a study showed elevated levels of lactate dehydrogenase (LDH) leakage and reactive oxygen species (ROS) generation in a GO concentration-dependent manner, indicative of mitochondrial oxidative stress, resulting in cell death [48]. Similarly, GO nanoplatelets caused dose- and time-dependent cytotoxicity in Hep G2 cells, with the induction of oxidative stress being the cause of toxicity [49]. Meanwhile, another study suggested that uniform ultrasmall GO (~50 nm) has high cellular uptake and low cytotoxicity [50]. Furthermore, Lv et al. showed that GO had no obvious cytotoxicity at low concentrations (<80 μg/mL), but the viability of neuroblastoma SH-SY5Y cells exhibited dose- and time-dependent decreases at high concentrations (≥80 μg/mL) [51]. Moreover, a study also showed that GO can induce oxidative stress and cause potential neurotoxicity with decreased neurotransmitter contents in *C. elegans* [52]. Interestingly, a recent study reported that adhesion states greatly affect cellular susceptibility to GO, showing that human HCT-116 cells in a non-adhered state are more susceptible to GO than those in an adherent state [53].

In terms of GO synthesis, various methods have been developed, with the improved or modified Hummers’ methods being the most commonly used [54,55,56]. Recently, we developed a method, using 1-pyrenebutyric acid (1-PBA) to facilitate GO synthesis by preventing graphite over-oxidation and facilitating GO separation, which allows for the preparation of different-sized GO, from <1 µm to >100 µm, with a higher quality compared with the commonly used methods [57].

In this study, we synthesized relatively small-sized GO (1 ± 0.4 µm) using the 1-PBA-assisted method [57]. Then, we assessed how this GO, as a filler, affects the mechanical properties of GO–gelatine nanocomposite films and the cytotoxicity of HEK-293 cells by growing the cells on the top of GO–gelatine substrates. Our results show that the addition of GO (0–2%) enhances the mechanical properties of the composites, from tensile strength and elasticity to ductility and toughness, in a GO concentration-dependent manner. The results of cell viability and a real-time phosphatidylserine-annexin V apoptosis–necrosis assays show no observable cytotoxicity for GO, as compared with pure gelatine. Furthermore, we performed circular dichroism (CD), X-ray diffraction (XRD), Fourier-transform infrared spectroscopy (FTIR), and X-ray photoelectron spectroscopy (XPS) analyses to elucidate the mechanism of the interactions between GO and gelatine. Our results reveal, for the first time, that GO enhances the mechanical properties of gelatine through the formation of non-covalent intermolecular interactions with gelatine at its amorphous or disordered regions.

## 2. Results and Discussion

### 2.1. Characterizations of GO Sample

In this study, GO nanosheets were prepared using the 1-PBA-assisted oxidation method, as described previously [57,58]. It was shown that 1-PBA can facilitate graphite separation and prevent its over-oxidation [57]. The GO was characterized using atomic force microscopy (AFM), which showed that the lateral sizes are 1 ± 0.4 µm, with thicknesses ranging between 1 and 2 nm, and about 91% of the GO are monolayer (Figure 1A,B, Table 1). The Raman spectra (Figure 1C) showed characteristic D and G bands associated with GO, and the I_D_/I_G_ intensity ratio was 0.93. Furthermore, the surface chemical characterization was analyzed using XPS (Figure 1D), which showed a very low degree of impurities (≤1%). The carbon-to-oxygen (C/O) ratio was 2.2, and, thus, the oxygen content of the GO was about 31%. High-resolution XPS spectra were also used to assess the content of various oxygenated functional groups of GO, and the results are summarized in Table 1.

### 2.2. Effects of GO on the Mechanical Property of Gelatine-GO Composite Films

Although gelatine has desirable biocompatibility properties, its limited mechanical strength hinders its applications within the biomedical field, like tissue engineering [59]. Therefore, we investigated how GO, as a filler, influences the mechanical properties of gelatine dry films. Using 0–2 wt% GO as fillers, free-standing GO–gelatine composite films were prepared using a vacuum filtration method for tensile strength testing. As shown in Figure 2, these films displayed very different tensile strength progress profiles (Figure 2A). Interestingly, all the tested GO–gelatine composites showed a higher tensile strength than that of the pure gelatine and GO films (Figure 2A). The tensile strength (the maximum stress that a material can withstand before breaking) of the composites increased in a GO concentration-dependent and non-linear manner (Figure 2B). The presence of 2 wt% GO increased the tensile strength of the gelatine films from 48.4 MPa to 149 MPa, about a 3.1-fold increase. The Young’s modulus (E) or the slope of the tensile stress–strain curve in the linear region represents the elastic modulus or stiffness of the material. It also increased with the increasing GO in the composites, with the presence of 2 wt% GO improving the elastic modulus from 20.8 MPa to 42.5 MPa (Figure 2C).

In addition, GO also elongated the strain of break that represents the ductility of the samples (Figure 2D). Whilst pure gelatine (3.7%) is more ductile than GO (1.23%), they are both brittle materials with a ductility of <5%. Interestingly, the ductility of all the composites increased to higher than 5%, thus becoming more ductile. For example, it increased from 3.7% of pure gelatine to 8.7% in the presence of 2% GO, about a 2.4-fold increase (Figure 2D). The area under the tensile stress–strain curve represents the amount of mechanical energy that a material can absorb before it breaks. This energy absorption capacity is often correlated with the material’s toughness, and a higher energy absorption capability implies a greater ability of a material to withstand forces resisting failure. Our results showed that the relative area under the stress–strain curve also increased with the increase in the GO % (Figure 2E), and the presence of 2% GO improved the energy absorption capacity of gelatine by about 8-fold. A similar mechanical enhancement result was shown with GO synthesized using a modified Hummers’ method in a recent study [60]. However, there, only the presence of 5% and 10% GO was studied. Whilst 5% GO achieved a similar result obtained with 1–2% GO in this study, 10% GO did not enhance the mechanical properties of the composite film further but decreased slightly [60].

Taken together, our results show that GO is an efficient nanofiller for enhancing the mechanical properties of gelatine. The addition of a small amount of GO (<2%) can effectively improve the mechanical properties of a gelatine film, from tensile strength and elasticity (stiffness) to ductility and toughness. Similar mechanical improvement results were reported previously, where GO was used as a nanofiller of synthetic polymers, like poly(methyl methacrylate) and polyvinyl alcohol (PVA) [61,62,63]. To our knowledge, this is the first report to show that all four key mechanical properties (strength, elasticity, ductility, and toughness) of gelatine are improved by GO. They are key mechanical properties used to characterize and evaluate materials. Such a mechanical property improvement using GO as a nanofiller is desirable for the wide potential applications of gelatine in tissue engineering and regenerative medicine [12]. For example, substrates with high tensile strength or stiffness can enable structural reorganization in response to the mechanical stress exerted by cells, such as during cell proliferation, which is a key requirement for tissue engineering and ECM modeling [64,65,66].

### 2.3. Effects of GO on the Cytotoxicity of HEK-293 Kidney Cells

To assess whether the GO synthesized using the 1-PBA-assisted method is biocompatible, we seeded HEK-293 kidney cells on top of gelatine, GO, or GO–gelatine composite-coated substrates. Firstly, gelatine, GO, or GO–gelatine was drop-casted and coated on glass cover slips within a 24-well plate, respectively. Then, 2 × 10^4^ cells were seeded on the top of these substrates (Figure 3A). After seeding, the cell attachment and morphology were regularly monitored using optical microscopy. Under all these conditions, the same characteristic cell morphology as that during cell culture in tissue culture flasks was observed. After the cells grew for five days, cell viability was analyzed using the Alamar blue assay, which showed no difference in the cell viability between these samples (Figure 3B). As a negative control, we tested and found that the cells cannot grow on glass without a coating.

Next, a more sensitive RealTime-Glo^TM^ phosphatidylserine-annexin V apoptosis and necrosis assay was performed. In this assay, apoptosis is measured by the luminescence intensity change resulting from the exposure of phosphatidylserine (PS) on the outer leaflet of the cell membrane during the apoptotic process, and necrosis is monitored based on the fluorescence resulting from the loss of membrane integrity during secondary necrosis at the same time [67]. This assay was performed upon the exposure of HEK-293 cells to the different substrates over a duration of 5 days (Figure 3C,D). The results showed that both the luminescence (apoptosis, Figure 3C) and fluorescence (necrosis, Figure 3D) signals increased with time initially, during the first 2–3 days, suggesting the presence of some cell apoptosis and necrosis in all samples initially. This may reflect the time required for appropriate cell attachment to the substrates since it was observed for all samples, and similar observations were reported previously for HEK-293 and other cell lines [67,68,69]. Importantly, the profiles were the same for all samples, confirming that the GO is not toxic to cells under these conditions.

Conflicting results for GO biocompatibility have been reported across the literature, as discussed in the Introduction, above. This is possibly due to several reasons, such as differences in GO purity, concentration, lateral size, incubation time, and cell types. For example, practically, GO samples may contain toxic residual chemicals from the synthesis process, such as liquid phase exfoliation and chemical oxidation. The residual contaminants can induce toxicity in cells and organs, as reported [45,46]. Therefore, it is no surprise that conflicting results about GO biocompatibility and cytotoxicity have been reported in the literature. The GO used in this study is dominated by monolayer sheets (>90%), with lower Raman I_D_/I_G_ ratios and higher C/O ratios compared with those prepared using the improved or modified Hummers’ methods [57]. A key difference between our current study and previous cytotoxicity studies is the method of GO incubation. We investigated the cytotoxicity of GO nanosheets by seeding cells directly on the surface of GO, whilst most previous studies [35,36,38,39,40,41,42,43,44,48,49] added GO in a solution or medium, in which the tested cells may or may not have direct interactions with GO. A previous study showed that GO is a good cell adhesion substrate [70]. Consistently, we found no difference in the cell attachment and shape under optical microscopy. Importantly, the GO has no apparent effects on cell viability, apoptosis, and necrosis for up to five days. Thus, our study using solid GO film (coating) provides new evidence to show that GO nanosheets have no apparent cytotoxicity upon direct interaction with HEK-293 cells. It would be interesting and important to investigate whether the GO synthesized using the 1-PBA-assisted method could induce oxidative stress and/or potential neurotoxicity under different conditions in the future.

### 2.4. Physicochemical Characterization of GO–gelatine Composites

A similar result of increased mechanical properties in a gelatine composite film was reported by [60] Layek et al. (2021). They performed FESEM, FTIR, Raman, and XRD analyses and concluded that the enhancements were obtained via the formation of an intercalated nanolaminate structure, hydrogen bonding interactions, and the tailoring of the crystal structure of gelatine in the composite film [60]. To understand the interaction between GO and gelatine further, complementary characterization techniques were used. First, to assess whether there is an effect of GO on the folding of gelatine, CD spectra of gelatine solution in the presence and absence of 1 wt% GO were recorded (Figure 4A). Both spectra showed a characteristic spectral signature of collagen-like coiled helices, with a negative peak at 200 nm [71]. The good overlap of the CD spectra suggests that no obvious folding or conformational change was induced by the presence of 1 wt% GO.

Next, X-ray diffraction (XRD), a versatile and non-destructive analytical technique, was used for the chemical and structural characterization of GO–gelatin films. XRD patterns of GO and GO–gelatine composite films are shown in Figure 4B. The GO spectrum showed a major characteristic (002) peak at 2*θ* = 9.9°, a characteristic peak of GO, corresponding to an interlayer spacing of 0.89 nm and, thus, monolayer GO. Gelatine exhibits two main diffraction peaks, with a small peak at a low-angle region of approximately 2*θ* = 8.4° arising from the amorphous or disordered regions of the gelatine and a more intense peak at 2*θ* = 20.8° arising from the ordered triple-helical structure [72]. In the GO–gelatine composite spectra, the GO peak (2*θ* = 9.9) was not apparent, which is likely due to the low GO concentration and an indication of good GO dispersion in the gelatine matrix [73,74]. Consistently, the gelatine peak at 2*θ* = 8.4° was slightly and gradually blue-shifted from 8.4 to 8.1 as the GO concentrations increased from 0 to 2 wt% (Figure 4B). The results suggest the presence of intermolecular interactions between GO and gelatine as well as changes in the molecular packing or arrangement of the gelatine structure. However, there were no obvious shifts for the 20.8° peak upon the addition of GO, indicating that the triple-helical structure of gelatine is not affected by GO.

In summary, CD and XRD analyses suggest that the interactions between GO and gelatine primarily impact the amorphous or disordered regions of the gelatine structure; no folded region and no apparent folding or structural change are induced.

To understand the interactions between GO and gelatine further, Fourier-transform infrared spectroscopy (FTIR) analysis was performed (Figure 4C). The presence of different oxygenated functional groups in GO was observed, as expected, which are C–O–C epoxy groups at 1213 cm^−1^, oxidized graphitic domain of GO at 1623 cm^−1^, and C=O stretching at 1723 cm^−1^ from carboxylic (COOH) groups [75]. The FTIR spectrum of gelatine shows characteristic bond vibrations at 1648 cm^−1^ from stretching of the C=O bond of amide I group, 1533 cm^−1^, representing the bending of N–H and stretching of C–N within amide II, and 1227 cm^−1^, corresponding to the stretching of C–N and bending of N–H from amide III groups [76,77]. Interestingly, in the composite spectra, all three peaks of gelatine (C=O, N–H, and C–N) shifted to lower wavelengths gradually with an increase in GO. They shifted from 1648 cm^−1^, 1533 cm^−1^, and 1227 cm^−1^ in pure gelatine to 1636 cm^−1^, 1523 cm^−1^, and 1220 cm^−1^, respectively, in the 2 wt% GO–gelatine composite (Figure 4C). These observations are clear evidence of the physical interactions between the GO and gelatine. A similar FTIR result was obtained previously using 5% and 10% GO [60]. The presence of interactions between GO and gelatine in the composite film was also confirmed by Raman analysis, where shifts in both the D band and G band of gelatine were observed due to the presence of GO, indicating the presence of interactions [60].

In addition, XPS (Figure 4D) was used to investigate the potential chemical interactions between GO and gelatine, as the positions of XPS peaks are sensitive to the chemical environment of the elements. The peaks at 284 eV, 400 eV, and 531 eV correspond to C1s, O1s, and N1s, respectively. The C/N ratio increased as the GO content increased in the nanocomposites, as expected (Table 2). The deconvolution results of the carbon (C1s) and nitrogen (N1s) spectra (Appendix A) are summarized in Table 2, which confirms that the relative percent of carbon–carbon bonds increased, while the various carbon–nitrogen bonds decreased as the percentage of GO increased. However, there is no observable peak shift for all composites, as compared with the pure gelatine, suggesting that no covalent bonds between GO and gelatine formed. Thus, the physical bonding between GO and gelatine is most likely non-covalent interactions, such as van der Waals forces, hydrogen bonding, and electrostatic interactions.

### 2.5. Mechanism for How GO Improves the Mechanical Property of Gelatine

As shown above, our CD, XRD, FTIR, and XPS characterizations revealed that the GO was well incorporated into the gelatine matrix via non-covalent intermolecular interactions, such as hydrogen bonding and hydrophobic and electrostatic interactions. Consistently, a similar conclusion was obtained previously with 5% GO [60] and during GO–gelatine hydrogel characterization [78]. Thus, the mechanical enhancement of gelatine by GO can be explained by the formation of the ammonium carboxylate complex via the interaction between the carboxyl groups of GO and amino groups of gelatine [79]. More importantly, in this study, we showed, for the first time, that GO strengthens the mechanical properties of gelatine through non-covalent interactions with its amorphous (disordered) regions, without an obvious effect on the folding of gelatine. This is consistent with the previous observation of thermogravimetric analysis (TGA) that, although the presence of GO (5% and 10%) altered the weight loss profile of gelatine, there was no obvious increase in or effect on the decomposition temperature of gelatine [60]. Taken together, we propose a model for how GO as a filler enhances the mechanical property of a GO–gelatine composite film.

As shown in Figure 5, non-covalent interactions, such as hydrogen bonding between the carboxylate and hydroxyl groups of GO and amino groups (on arginine and lysine) or hydroxyl groups (on serine, threonine, and hydroxyproline) of gelatine, are formed. GO acts like a non-covalent crosslinker hub, allowing a network of non-covalent interactions to form between the negatively charged GO and overall positively charged gelatine (pI 7–9.5). Apart from hydrogen bonds, the interactions may also include van der Waals forces and hydrophobic and electrostatic interactions, but since gelatin lacks aromatic residues, the potential π-π stacking interaction is less likely. Such a network of non-covalent interactions improves the mechanical properties of GO–gelatine composites efficiently. The larger surface area and multiple crosslinking sites on each GO nanosheet lead to a better stress transfer from gelatine to GO and between different gelatine chains than that in the absence of GO. Thus, GO renders GO–gelatine films with high mechanical performance. Moreover, the interactions between GO and gelatine primarily impact the disordered regions of the gelatine, without an observable effect on the folding and conformation of gelatine. GO not only reinforces the tensile strength and elasticity but also ductility and toughness of GO–gelatine composite films, making the composite’s resistance to deformation greater compared to gelatine alone. Such a composite could be a potential material of choice for use in biomedical engineering. We hope that our findings may help to pave the way for potential and wide applications of GO in tissue engineering and regenerative biomedicine.

## 3. Materials and Methods

### 3.1. Materials

Graphite flakes (Sigma Aldrich, >80% 100-mesh), potassium permanganate (KMnO_4_) (>99.5%), gelatine powder (type A, porcine skin, pI 7–9.5, and Mw range: 50–100 kDa) were purchased from Sigma-Aldrich (Gillingham, UK). Concentrated sulfuric acid (H_2_SO_4_, 95%), DMEM/F12 Ham media, fetal bovine serum (FBS), penicillin-streptomycin trypsin-EDTA (0.05%), and trypan blue (0.4%) were from Fisher Scientific (Loughborough UK). 1-pyrenebutyric acid (1-PBA, 97%) and 30% hydrogen peroxide (H_2_O_2_) were purchased from Alfa Aesar (Heysham, UK). Resazurin (Alamar blue reagent) and RealTime-Glo^TM^ annexin V apoptosis–necrosis assay were obtained from Promega (Southampton, UK). All chemicals were of analytical grade and used as received. Human HEK-293 epithelial kidney cells (CRL-1573) were a gift from Prof Tao Wang (University of Manchester) and purchased from ATCC, (Manassas, VA, USA).

### 3.2. GO Synthesis

GO was synthesized using 1-pyrenebutyric acid-assisted method [57]. Briefly, 100 mg 1-pyrenebutyric acid (1-PBA) was added to 50 mL of concentrated sulfuric acid (95%) and stirred for 1 h before the addition of 1 g of graphite flakes into the mixture under 10 min of mechanical stirring. The 1-PBA-graphite suspension was then left to stand for 2 days at room temperature. Prepare the oxidizing agents in a separate mixture, 10 g of KMnO_4_ was added to 50 mL of 95% sulfuric acid with mechanical stirring for 30 min. The oxidizing agents were then added slowly to the 1-PBA-graphite mixture under mechanical stirring for 10 min and left to stand overnight at room temperature. After the overnight oxidation, the mixture was placed in an ice-bath, and 400 mL cold MQ water was added slowly to the mixture to dilute the acidic solution. Then, to remove the unreacted purple paste of KMnO_4_, approximately 6 mL 30% H_2_O_2_ was added gradually into the mixture until no change or effervescence was observed. The supernatant was carefully drafted off, then 1 L MQ water was added to wash the GOs, and this process was repeated at least 5–6 times until pH of the sample increased to about 6.

### 3.3. Preparation of GO, Gelatine, and GO–gelatine Nanocomposite-Coated Substrates for Biocompatibility Studies

For biocompatibility, two starting stock solutions, (A1) 1 mg/mL gelatine and (B1) 1 mg/mL GO suspension in PBS, were prepared first. Gelatine was dissolved in PBS at 45 °C for 15 min in water bath to form 1 mg/mL gelatine aqueous solution. GO–gelatine samples of different GO concentrations (0–2 wt%) were prepared by mixing the stock solutions, A1 and B1, with different volumes. For example, to prepare 1% gelatine-GO solution, 0.3 mL of 1 mg/mL GO (B1) was mixed with 29.7 mL of the gelatine stock solution (A1), vortexed briefly, and sonicated for 15 min at 45 °C in a water bath. Subsequently, 40 µL of solution was drop-casted onto each circular glass cover slip within 24-well tissue culture plate. They were left to dry in an incubator hood overnight under UV for sterilization.

### 3.4. Preparation of Gelatine and GO–gelatine Nanocomposite Films for Material Characterisations

For mechanical strength testing, large films of 47 mm diameter were prepared using 10 mL of solutions, as required. Two starting stock solutions, (A100) 100 mg/mL gelatine and (B1) 1 mg/mL GO suspension, were prepared first, similar to that described in Section 2.3. Then, various GO–gelatine solutions were prepared by mixing the stock solutions A100 and B1 with different volumes accordingly. For instance, to prepare 1% gelatine-GO solution, 1 mL of 1 mg/mL GO (B1) was mixed with 2.97 mL of 100 mg/mL gelatine (A100), followed by dilution with PBA to 10 mL; then, the mixture was vortexed and sonicated for 15 min at 45 °C using a water bath. The solutions were vacuum filtrated on hydrophilic polytetrafluoroethylene (PTFE) membrane (Millipore 0.2 μm pore size), for approximately 2 days until all aqueous solutions were not visible. Subsequently, 70% ethanol was used to help peel off the dried films from the PTFE membrane. The film thickness was measured using SEM imaging as 30 µm [58].

### 3.5. Cell Culture

Human HEK-293 epithelial kidney cells (ATCC, CCL-1573) were maintained and passaged in DMEM/F12 Ham media supplemented with 10% FBS (Fetal Bovine Serum, 1% of penicillin/streptomycin), at 37 °C in 5% CO_2_. Cells were passaged using 0.05% trypsin-EDTA, when reaching 80–90% confluence. Typically, cells were seeded onto various dried sterilized GO or gelatine substrates coated on the top of round cover slips (13 mm, Academy), within 24-well plates (20,000 cells/well). Then, cells were left to adhere and proliferate for 5 days or, as indicated, at 37 °C in the CO_2_ incubator before a cell viability assay was performed. After seeding, the cell attachment and morphology were regularly monitored using optical microscopy.

### 3.6. Cell Viability Assay

Cell viability in terms of mitochondrial integrity and overall cellular metabolism was measured using Alamar blue assay, which uses cell membrane-permeable reagent resazurin, a blue and non-fluorescent reagent [80]. After 5 days incubation of cells on the various GO/gelatine substrates, resazurin (40 µL) was added to 400 µL (1:10 dilution) of culture medium in each sample well of a 24-well plate. The solution was mixed and incubated for 60 min in the dark at 37 °C and 5% CO_2_. Then, 100 µL of solution in each well was transferred to a well of 96-well plate, and the absorption intensities at 570 and 600 nm were measured using a plate reader BioTek Synergy H1 (Loughborough, UK). The cell viability (%) is represented with cells grown on the gelatine substrate as 100%. As a negative test, negligible cell growth was observed on the non-coated glass. Statistical analyses were performed using an unpaired Student’s *t*-test in GraphPad PRISM 8.0.0 (GraphPad Software, San Diego, CA, USA). At least 3 independent experiments were performed under each condition.

### 3.7. RealTime-Glo^TM^ Annexin V Apoptosis-Necrosis Assay

We used a live cell real-time assay that measures the exposure of phosphatidylserine (PS) on the outer leaflet of the cell membrane during apoptotic process. In this assay, a luminescent probe (annexin V) is used for detecting apoptosis as it can specifically bind to phosphatidylserine (PS), a phospholipid normally located on the inner leaflet of the plasma membrane but exposed on the outer leaflet during the early stages of apoptosis. Meanwhile, the presence of a pro-fluorescent DNA detection dye is used to measure necrosis for the loss of membrane integrity, allowing the dye to translocate to nucleus interacting with DNA and produce fluorescent signals. In this study, the reagents were prepared according to the manufacturer’s protocol (Promega, Cat#: JA1011). The assay reagents (25 µL/well) were added to the initial cell-laden plate (20,000 cells/well) and incubated within the cell culture in incubator until measurement was needed on each day for a time course of 5 days. The PS translocation and, thus, apoptosis were measured based on relative luminescence intensity/unit increase (RLU). The membrane integrity and, thus, cell necrosis were measured based on relative fluorescence intensity/unit change (RFU), with excitation and emission wavelengths at 485 ± 10 nm and 525 ± 10 nm, respectively. The data were analyzed based on the pseudo-first-order association kinetics, calculated using:(1)$$Y=Y0+\left(P-Y0\right)\times (1-\exp\left[-kX\right])$$
where Y represents RLU or RFU, Y0 is the initial value, P is plateau, which is the value at infinite times, k is the rate constant (day^−1^), and X is the time.

### 3.8. Circular Dichroism (CD) Measurement

CD samples of 1 mg/mL gelatine and gelatine–GO (1 wt%) solutions were prepared by dilution from starting solution of 29.7 mg/mL stock solutions with PBS. Then, 100 µL of these solutions was pipetted in to avoid formation of air bubbles. The spectra were acquired in a wavelength range 190–260 nm with Applied Photophysics Chirascan spectrometer (Leatherhead, UK) with a 0.1 mm quartz cuvette at 20 °C, at 0.5 s per point, and 1 nm bandwidth interval. The spectra were background subtracted using PBS or 0.01 mg/mL GO in PBS as blank for gelatine and GO–gelatine samples, respectively.

### 3.9. Physicochemical Material Characterizations of GO/Gelatine Films

Atomic force microscopy (AFM) Bruker Multimode 8 (Coventry, UK) measurements. Nanoscope Analysis Software 1.7 (Coventry, UK) was used to process the obtained AFM images and to calculate histograms of sheet thickness and lateral size.

X-ray diffraction (XRD) was performed with a PANalytical X’Pert Pro X’Celerator (Manchester, UK) diffractometer with Cu Kα radiation (λ = 1.5406 Å) at step size of 0.02° and 5 s per step. Interlayer spacing of GO was calculated using Bragg’s law, λ = 2d sin(*θ*), where λ is X-ray wavelength, d is the interlayer spacing of GO, and *θ* is the diffraction angle.

Fourier-transform infrared spectroscopy (FTIR) measurements were performed on a Bruker Alpha-P FTIR (Coventry, UK) spectrometer with a diamond ATR accessory. Spectra were obtained at a resolution of 4 cm^−1^ and 128 scans, with respect to IPA solution background. The FTIR test was conducted in a typical room-temperature and pressure environment. The GO and GO–gelatine samples were then scanned over several angles of incidence, and the refracted beam was detected to give the spectra across 1000–2000 cm^−1^.

X-ray photoelectron spectroscopy (XPS) was carried out using an Axis Ultra Hybrid spectrometer (Kratos Analytical, Manchester, UK), using monochromatic Al Kα X-ray radiation at 1486 eV (10 mA emission at 15 kV, 150 W), under ultra-high vacuum at a base pressure of 1 × 10^−8^ mbar. GO, gelatine and GO–gelatine samples were pressed onto conductive copper tape. A charge neutralizer was used to remove any differential charging at the sample surface. Calibration of the binding energy (BE) scale was performed using the C1s photoelectron peak at ~284.6 eV for graphitic carbon. High-resolution C1s and N1s spectral deconvolution was performed using CASAXPS software 2.3.26 (Casa Software Ltd., Devon, UK) with Shirley-type background subtraction. Graphitic carbon was fitted with an asymmetric peak shape, analogous to those used to fit metallic peaks, due to the high conductivity of this carbon species. Graphitic carbon also exhibits a signature broad spectral feature between approximately 290 and 295 eV (a BE region where no other carbon species are expected), associated with excitation of the π-π* transition, which was fitted with one broad peak. Gaussian–Lorentzian functions were fitted to all identified functional groups, which are constrained to the following binding energies: C–C and C=C at 284.5–284.6 eV; C–N at 286–286.4 eV (C1s) and 399–400 eV (N1s); O–C=O/O=C–N at 288.6–290 eV (C1s) and C=N at 400.2–401 eV.

The tensile strengths of all films (5 mm × 25 mm dumbbell-shaped strips) of 30 µm thickness were measured using an Instron 3342 universal testing machine (High Wycombe, UK) at a loading rate of 0.5 mm min^−1^ with a gage length of 10 mm. Young’s modulus of gelatine, GO, and GO–gelatine composite films was calculated based on:(2)$${Young}^{’}smodulus=\frac{\sigma }{\epsilon }$$
where σ represents tension stress (MPa) and ϵ is axial strain, according to Hooke’s law of elasticity [81].

### 3.10. Statistical Analysis

Data are represented as mean ± standard error of mean (SEM). Where relevant, statistical analyses were performed using one-way ANOVA, GraphPad PRISM 8.0.0 (GraphPad Software, San Diego, CA, USA).

## 4. Conclusions

In this study, we showed how GO (1.0 ± 0.4 µm) synthesized using the 1-PBA-assisted method has good biocompatibility based on the cytotoxicity analysis of HEK-293 cells grown on GO–gelatine-coated substrates. We also showed that the addition of GO as fillers (0–2%) improves the mechanical properties of gelatine in a concentration-dependent manner. The presence of 2 wt% GO increased the tensile strength, elasticity, ductility, and toughness of the gelatine films by about 3.1-, 2.5-, 2-, and 8-fold, respectively. CD, XRD, FTIR, and XRPS analyses allowed us to conclude, for the first time, that GO strengthens the mechanical properties of gelatine through non-covalent interactions with its amorphous or disordered regions. We believe that our findings will provide new insight and help pave the way for potential and wide applications of GO in tissue engineering and regenerative biomedicine.

## Figures and Tables

**Figure 1 molecules-29-02700-f001:**
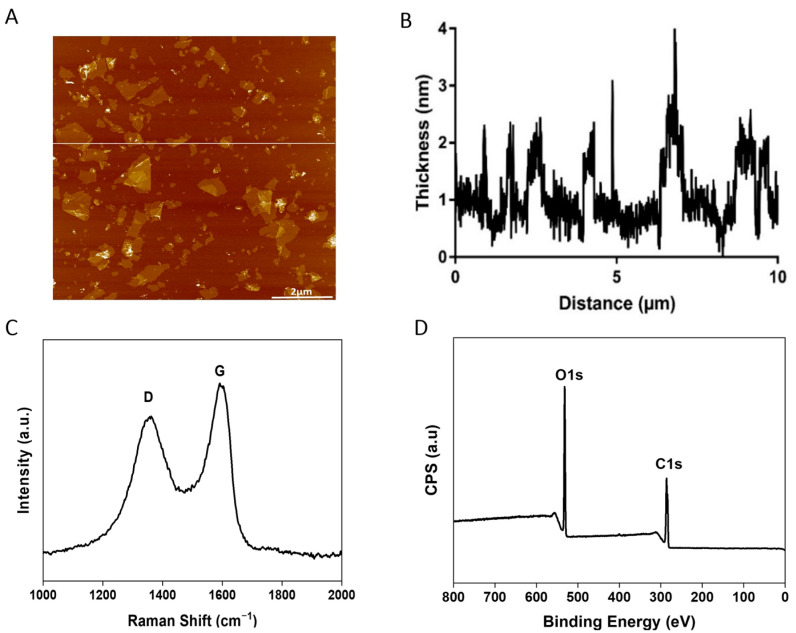
Characterizations of GO used in this study. (**A**,**B**) AFM image and corresponding height profiles (**C**) Raman spectrum; (**D**) XPS survey spectrum. More detailed information can be found in Table 1 and our previous publications [57,58].

**Figure 2 molecules-29-02700-f002:**
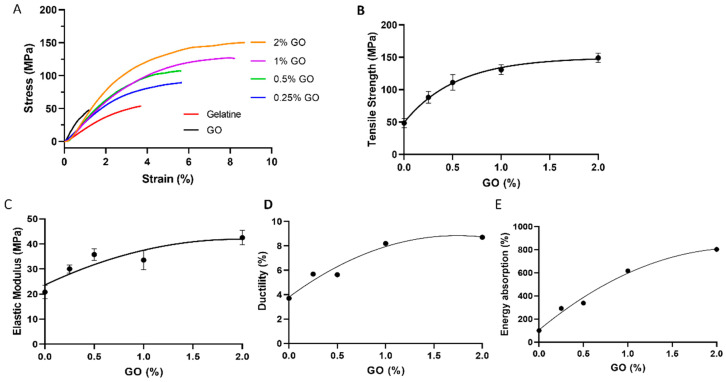
Mechanical properties of gelatine, GO, and GO–gelatine nanocomposite films. (**A**) Stress–strain tensile behaviour profiles with the color representations indicated. (**B**–**E**) Correlations between GO (wt%) and the tensile strength (**B**), elastic modulus (**C**), ductility (**D**), and relative energy absorption (**E**). The error bars represent the standard error of the mean (SEM) of three independent experiments (*n* = 3) for all samples.

**Figure 3 molecules-29-02700-f003:**
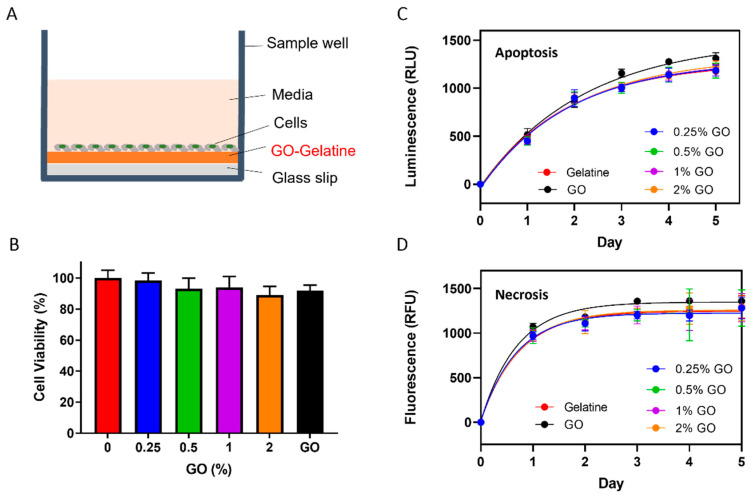
Assessment of cell viability on substrates coated with gelatine, GO, and various GO–gelatine composites. (**A**) Cross-sectional schematic diagram of the experimental set-up in a single well of a 24-well plate. (**B**) Cell viability based on Alamar blue assay after the cells were grown in different substrates for 5 days. (**C**,**D**) Time courses of RealTime-Glo^TM^ phosphatidylserine-annexin V apoptosis (**C**, Luminescence) and necrosis (**D**, Fluorescence) assays. The solid lines represent pseudo-first-order association kinetics. The error bars represent the SEM of at least three independent experiments (*n* ≥ 3) for all samples. One-way ANOVA showed no statistically significant difference in all cases, *p*-values > 0.05.

**Figure 4 molecules-29-02700-f004:**
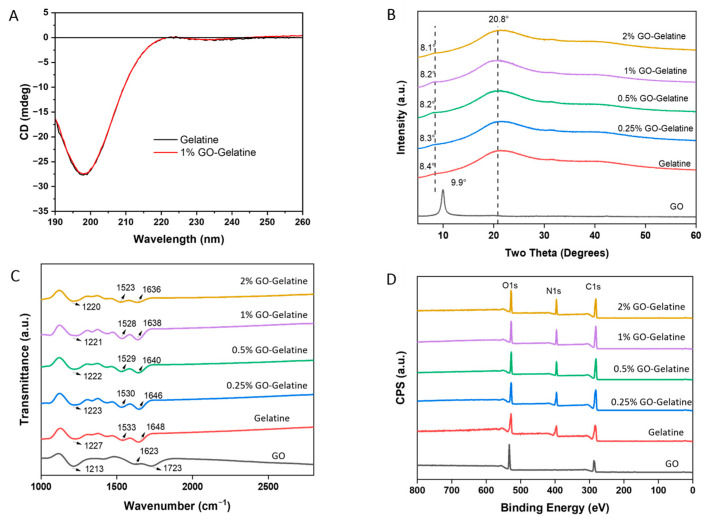
Physical and chemical characterizations of GO–gelatine composites. (**A**) Far UV CD spectra of gelatine solution in the absence and presence of 1 wt% GO. (**B**) XRD pattern; (**C**) FTIR spectra; and (**D**) XPS survey spectra of various GO–gelatine composite films. The color representations are the same in (**B**–**D**), as indicated in the plots.

**Figure 5 molecules-29-02700-f005:**
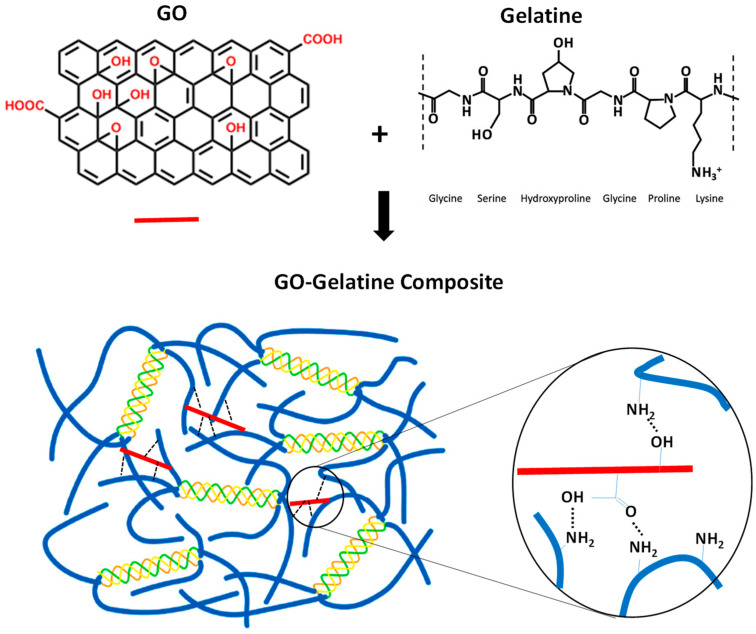
Schematic diagram of how GO improves mechanical property of GO–gelatine nanocomposite. Basic chemical structures of GO and gelatine are presented on top row. The proposed GO–gelatine composite interactions are shown in the bottom row, with examples of intermolecular hydrogen bonding interactions (black dash lines) formed between GO (red) and gelatine chain at amorphous or disordered ranges (blue) dispersed on the right. The triple-helix structure of gelatine is shown in three colors.

**Table 1 molecules-29-02700-t001:** A summary of the key physical and chemical properties of GO nanosheets.

Technique	Property	Value
AFM	Lateral Size	1.0 ± 0.4 µm
	Flake Thickness	1–2 nm
	Monolayer	91%
Raman	I_D_/I_G_ ratio	0.93 ± 0.1
XPS	Purity (%)	99
	C/O ratio	2.2 ± 0.1
XPS C1s Deconvolution	C=C/C–C	33.8%
	C–O	42.9%
	O–C=O	21.4%
	π-π*	1.9%

**Table 2 molecules-29-02700-t002:** Summary of XPS analyses of gelatine and various GO–gelatine nanocomposites. Chemical composition measured by high-resolution XPS. Error bars represent standard error of mean (SEM), *n* = 3.

	C/N Ratio	C=C/C–C (%)	C=N (%)	C–N (%)	O=C–N (%)
Gelatine	2.02 ± 0.06	32 ± 3	26 ± 3	38 ± 2	4 ± 1
0.25% GO–gelatine	2.20 ± 0.05	34 ± 1	26 ± 2	37 ± 3	3 ± 1
0.5% GO–gelatine	2.24 ± 0.03	35 ± 2	26 ± 2	38 ± 1	2 ± 1
1% GO–gelatine	2.32 ± 0.01	36 ± 1	24 ± 1	38 ± 3	2 ± 2
2% GO–gelatine	2.49 ± 0.03	38 ± 2	23 ± 2	37 ± 4	2 ± 2

## Data Availability

The datasets generated during and/or analyzed during the current study are available from the corresponding author on reasonable request.

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
