# Peer review of "Graphene Oxide Strengthens Gelatine through Non-Covalent Interactions with Its Amorphous Region"

_molecules, 2024, doi:10.3390/molecules29112700_

Round 1

Reviewer 1 Report

Comments and Suggestions for Authors

The manuscript by Sim et al is an interesting work on the use of GO for improving mechanical integrity of gelatine. The paper can be accepted after very minor revisions. I would suggest some change in the title (I would avoid the use "covenant" instead of "covalent", for instance). Also I expect the authors to provide some details on the status (dry or wet) of the gelatine and gelatine-GO nanocomposites as it is not clear. Overall it is a nice work.

Comments on the Quality of English Language

Mostly ok.

Author Response

Thank you very much for your very positive comments and appreciation of our manuscript. Thank you for careful reading and identifying the typo mistake, and we changed ‘covenant’ to ‘covalent’ in the revised manuscript. We also explained it was ‘dry films’ at lines 13 and 146, respectively.

Reviewer 2 Report

Comments and Suggestions for Authors

Comments from Reviewer

Graphene oxide strengthens gelatine through non-covenant interactions with its amorphous region

The current form's presentation of methods and scientific results is unsatisfactory for publication in the Molecules journal. Some comments apply to the entire article. Please take this into account when making corrections. The minor and significant drawbacks to be addressed can be specified as follows:
1.    Line 24. Graphene Oxide ---> Graphene oxide
2.    Some abbreviations are explained several times in the text. It is sufficient to do it the first time. For example, "Circular Dichroism (CD)" – lines 97 and 250.
3.    Fig. 4. ProszÄ™ użyć takich samych kolorów dla poszczególnych próbek jak na poprzednich rysunkach
4.    Fig. 4(b). SGO?
5.    Fig. 5. The black dash lines are invisible – bottom – left panel.
6.    Line 403. Cell Culture ---> Cell culture.
7.    Line 449. Circular Dichroism ---> Circular dichroism
8.    Literature should also be standardized: the size of letters in the titles of journals, initials of names, the size of letters in the titles of articles. References require further format modification and refinement. See for example (i) Simultaneous feeding of volume and volume is not necessary - line 689: "Carbon 2006, 44, (15), 3342-3347" and line 609: "Carbon 2018, 129, 428-437." (ii) lowercase/uppercase letter at the beginning of a word - line 603: "of doxorubicin loaded graphene quantum" and line 613: "Surface Oxidation of Graphene Oxide Determines" (iii) line 661 "In Polymers, 2022; Vol. 14."????? ---> Polymers 2022, 14, 1548
9.    Conclusions?
10.    If I believe correctly, the best results were obtained for the 2% GO sample. Did the authors try to increase the content to 3% or even 5%? Results?

Sincerely,
    The reviewer.

Reviewer 3 Report

Comments and Suggestions for Authors

Reviewer comments:

1-            The SEM, TEM, and TGA of GO and gelatine-GO composites should be prepared.

2-            This work should be compared with similar results in the literature.

3-            The novelty of this work should be highlighted.

4-            What is the role of 1-pyrenebutyric acid (1-PBA) in the fabrication of GO?

5- The quality of Table s1 should be improved.

6-Auther should explain the possible mechanism and interaction between functional groups on the surface of GO and Gelatin.

7-Auther should explain that changing the percentage of GO, how can be detected by TGA characterization and Raman.

Round 2

Reviewer 2 Report

Comments and Suggestions for Authors

The authors revised their manuscript well. The revised version can be accepted for publication.

Reviewer 3 Report

Comments and Suggestions for Authors

Dear Editor of Molecules,

I hope this message finds you well.

I have reviewed the revised manuscript entitled " Graphene oxide strengthens gelatine through non-covenant interactions with its amorphous region" submitted to Molecules. The authors have addressed all the comments and suggestions provided during the review process comprehensively and satisfactorily.

Based on the thorough revisions and improvements made, I am confident that the manuscript now meets the high standards required for publication. Therefore, I am pleased to recommend that this paper be accepted for publication in Molecules.

Thank you for considering my recommendation.

Best regards,